# Enhancing Text-to-SQL Capabilities of Large Language Models: A Study on Prompt Design Strategies

**Linyong Nan**[1]    **Yilun Zhao**[1]    **Weijin Zou**[1]    **Narutatsu Ri**[2]    **Jaesung Tae**[1]
**Ellen Zhang**[1]    **Arman Cohan**[1,3]    **Dragomir Radev**[1]

[1]Yale University    [2]Columbia University    [3]Allen Institute for AI
{linyong.nan, yilun.zhao}@yale.edu

## Abstract

In-context learning (ICL) has emerged as a new approach to various natural language processing tasks, utilizing large language models (LLMs) to make predictions based on context that has been supplemented with a few examples or task-specific instructions. In this paper, we aim to extend this method to question answering tasks that utilize structured knowledge sources, and improve Text-to-SQL systems by exploring various prompt design strategies for employing LLMs. We conduct a systematic investigation into different demonstration selection methods and optimal instruction formats for prompting LLMs in the Text-to-SQL task. Our approach involves leveraging the syntactic structure of an example's SQL query to retrieve demonstrations, and we demonstrate that pursuing both diversity and similarity in demonstration selection leads to enhanced performance. Furthermore, we show that LLMs benefit from database-related knowledge augmentations. Our most effective strategy outperforms the state-of-the-art system by 2.5 points (Execution Accuracy) and the best fine-tuned system by 5.1 points on the Spider dataset. These results highlight the effectiveness of our approach in adapting LLMs to the Text-to-SQL task, and we present an analysis of the factors contributing to the success of our strategy.

## 1 Introduction

Question answering using structured knowledge source is a critical function of information retrieval systems that act as an interface between humans and vast structured data repositories. Extracting and aggregating information accurately is a fundamental requirement of these systems and is thus a primary goal in their design. In recent years, the neural symbolic design approach (Berant et al., 2013; Yao and Van Durme, 2014; Liang et al., 2017; Gardner et al., 2018; Yu et al., 2018; Cheng et al., 2023) has become the preferred choice for such

systems for two main reasons. First, neural models have inherent limitations, including a limited working memory that is costly to access during inference and a long-term memory that is unreliable to read from or write to, making it impractical to have them directly read from large-scale knowledge sources. Second, understanding how a system decides which information to retrieve and how to aggregate it is crucial for assessing its reliability and robustness.

Recent investigations have demonstrated the effectiveness of the neural symbolic approach in producing transparent reasoning process in formal language sequence (such as Text-to-SQL) for question answering tasks based on databases or knowledge graphs (Berant et al., 2013; Zhong et al., 2017; Yu et al., 2018; Yin and Neubig, 2018; Yu et al., 2019; Ren et al., 2021; Cheng et al., 2023). A typical system comprises a neural semantic parsing module that translates user queries in natural language to formal language sequences (e.g., logical forms or executable code) and a symbolic reasoner module, such as database management system (DBMS), that executes the code on structured knowledge sources to extract the result. The primary objective of this work is to improve the semantic parsing module, as it is essential in extracting answers from relational databases using SQL as the formal language.

Current semantic parsing modules can be broadly categorized based on their learning strategies. State-of-the-art systems involve fine-tuning a pretrained language models on a large corpus of {*question*, *SQL*} pairs, enabling the model to generate code (Wang et al., 2020; Yin et al., 2020; Scholak et al., 2021; Xie et al., 2022; Li et al., 2023). Alternatively, the in-context learning (ICL) approach exploits the inherent capabilities of large language models (LLMs) to directly produce SQL code by providing a well-defined task prompt (Xie et al., 2022; Chen et al., 2022; Rajkumar et al., 2022; Ni et al., 2023). Existing research indicates

that LLMs using prompt-based semantic parsing underperform their fine-tuned counterparts (Liu et al., 2023), while recent studies also suggest that performance of ICL-trained LLMs is significantly affected by the structure of the prompt (Liu et al., 2022; Rubin et al., 2022; Lu et al., 2022; Wei et al., 2022; Fu et al., 2023; Ye et al., 2023). This motivates us to examine various prompt configurations for semantic parsing tasks, taking advantage of the latest advancements of LLMs pertaining to our domain of interest.

Our study focused on exploring various prompt design strategies for semantic parsing tasks in the Text-to-SQL domain. We conducted a systematic investigation into different demonstration example selection criteria and instruction formats on Text-to-SQL datasets. Specifically, we propose to employ an example's SQL syntactic structure as the basis for retrieving demonstrations, thereby facilitating a more accurate representation of the problem structure. Our approach revealed that selecting demonstration examples with a dual emphasis on diversity and similarity objectives yields maximized gain in performance. Our study also showed that LLMs benefit from database-related knowledge augmentation in certain circumstances. Through experiments, we identified the most effective strategy, which resulted in an Execution Accuracy score of **84.4** on the Spider dataset (Yu et al., 2018). This score is **2.5** points higher than the state-of-the-art system (Ni et al., 2023) and **5.1** points higher than the best fine-tuned system (Scholak et al., 2021) at the time of writing.[1] These results demonstrate the effectiveness of our in-context learning scheme in adapting LLMs to our target task. Furthermore, we present the empirical findings and analysis on the factors that contributed to the success of our strategy.[2]

## 2 Methods

To design prompts for in-context learning in zero-shot or few-shot settings, it is important to find an optimal way to represent, augment, and arrange all resources in the input-output mapping. Additionally, the task instructions should be formulated to align with these resources. When few-shot learning is employed, the selection of a subset of demon-

strations from a pool of annotated examples for each test instance is another critical design choice that can impact the ICL performance. We proposed enhancements for each of these components and evaluated them against existing methods.

### 2.1 Demonstration Selection

The goal is to select a subset of annotated examples from a pool that offers the best context for solving the test problem. While random selection from the pool is one option, Liu et al. (2022) proposed $k$NN-augmented example selection (KATE), which retrieves $k$ nearest neighbors from the pool based on the input of the compared instances. To achieve this, all the pool instances are first transformed into continuous vectors using a sentence encoder. During inference, the input of a test instance is projected into a latent space using the same encoder and then compared to the pool of vectors using a similarity measure, such as negative Euclidean distance or cosine similarity. Finally, the top $k$ most similar annotated examples are selected from the pool.

**Structured Prediction as Basis for Retrieval** We propose utilizing the output SQL queries to select the demonstration examples, rather than using the input questions. This is because, unlike many tasks where the output is a classification label or extracted entity with little information about the problem structure, Text-to-SQL demands structured prediction which contains more explicit information about the problem structure than that provided in the input question. Furthermore, unlike natural language questions that can only be converted into continuous semantic vectors, SQL queries can be easily transformed into discrete feature vectors based on their syntax, making their comparison more efficient and transparent. To implement our proposal, we begin by converting the SQL queries of all pool instances into discrete syntax vectors. This is done by parsing the queries and identifying their syntactic elements, including keywords, operators, and identifiers. Each SQL query is then mapped to a "Bag-of-Syntactic-Elements" feature vector, each entry of which indicates the presence of a syntactic element in the query, i.e., we assign 1 (instead of the count) if an element is present in the SQL query. During inference, we first generate a draft of the SQL query using a preliminary predictor. We then apply the same process to convert this draft query into a discrete

---

[1]Our comparison focuses on fine-tuning studies that employed the standard Transformer architecture without any layer modifications or the inclusion of additional modules.

[2]We will open source our code for experiments: `https://anonymous.url`

vector, which is used to represent the test instance for retrieving demonstration examples.

**Balancing Diversity and Similarity** We propose a new demonstration selection strategy that differs from Liu et al. (2022), which retrieves the most similar examples with continuous-valued measurements for each test instance. In contrast, our strategy seeks to balance similarity and diversity of the demonstrations. This is achieved by changing the representation of the given example from a continuous-valued vector denoting the question semantics to a discrete-valued vector that captures the SQL syntax. To obtain demonstration examples that are similar to the given example, we first split the pool of annotated examples into disjoint partitions that represent different categories. Specifically, we use the difficulty-level based categorization derived from the Spider dataset (Yu et al., 2018), because it is developed strictly based on syntactic coverage and structure of a SQL query, ensuring that queries satisfying the same conditions are grouped into the same category. While alternative categorization options may exist, we leave this for exploration in future work. Given a test instance, we use a preliminary predictor to generate a draft SQL query and, based on its category, retrieve candidate examples that belong to the relevant partition. Next, to select diverse examples from the candidate partitions, we implement $k$-means clustering on the discrete vectors of examples, selecting $k$ diverse examples that are closest to each centroid of the cluster. The resulting examples exhibit similarity to the test example by sharing the same category, yet maintain diversity in problem structures. These demonstrations are then used to construct the prompt. The procedure for our demonstration selection strategy is outlined in Algorithm 1 of the appendix.

## 2.2 Schema Representation in Instruction

Instructions are crucial to designing prompts, as they define the task by clarifying how provided resources can aid the inference process (Dong et al., 2023). Our primary focus lies in determining the optimal way to represent a structured knowledge source within the instruction and identifying supplementary resources that can enhance the inference process.

**Linearization of Structured Knowledge** We begin by altering the linearization of structured knowledge. In prior research (Xie et al., 2022), structured knowledge sources such as databases or tables have been linearized into a "text" sequence. Building on previous methods (Rajkumar et al., 2022), we adopt a representation of the database using a "code" sequence, specifically the CREATE query employed to construct the table initially, as illustrated in listing 1 and 2 of the Appendix. This linearization approach provides data type information for each column and encompasses all foreign key constraint details within the database. Moreover, we modify other resources in the instructions, such as the question and example entries in the database, to conform to the code sequence style by appending them as comments.

**Schema-related Knowledge Augmentation** The ontology of a database delineates the structure and semantics of the database by offering definitions for a set of classes (tables), their attributes (columns), and the relationships among them. We initially enhance the semantics of each class and attribute by elaborating on their meanings within the context of the entire database. Specifically, we employ OpenAI's gpt-3.5-turbo engine[3] to generate a natural language definition for each column in every table, considering all its values and other columns. We then incorporate these definitions into the input either by appending them as a block comment or inserting them within the CREATE query as inline comments. Furthermore, we suggest augmenting the representation of the database structure by providing an Entity-Relationship summary that outlines the connections between tables and specifies how they can be joined. As depicted in Figure 9 of the Appendix, an Entity-Relationship diagram of a database is utilized to enumerate all possible paths between distinct tables. These paths are subsequently arranged in descending order based on their respective lengths. The resulting summary has shown to be useful in our experiments for test instances where multiple tables need to be combined. Listing 5 further demonstrates our augmentations and how we arrange them to construct the prompt.

## 2.3 Integrated Strategy for Text-to-SQL

Upon examination, we found that models trained with ICL exhibit sensitivity to the number of demonstration examples, resulting in noticeable variance in performance across models provided

---

[3]Public API available at https://openai.com/api/.

with various numbers of demonstrations. To establish substantial conclusions when comparing distinct prompting approaches, we present the mean and standard deviation for models sharing identical configurations except for the varying number of demonstrations. In addition, we employ a majority vote on these models exhibiting diverse performances. Specifically, we obtain the execution results of different models' greedy decoding predictions, eliminate those with execution errors by deterministic database management system (DBMS), and choose the prediction that receives the majority vote. Alternative integration methods, such as the self-consistency sampling (Wang et al., 2023), are also available, but we reserve their exploration for future research. The comprehensive results are available in Figures 10, 11, 12 of the Appendix for reader's perusal.

We propose the following procedure for constructing prompts for the Text-to-SQL task. Given a set $A$ of annotated examples, we first establish a categorization that divides the pool into disjoint partitions $A^\alpha, A^\beta, \ldots,$, with each partition containing examples whose SQL queries share a relatively similar syntax structure. Next, we apply the $k$-Means strategy detailed in Section 2.1 to obtain diverse demonstration examples $D^j$ for partition $A^j$. For each example, the demonstration is constructed by transforming the database into multiple CREATE queries and augmenting with schema-related knowledge. During inference, we employ a preliminary model to generate a draft SQL query, which is used to determine the problem category and thus the corresponding $D^j$ for building the prompt. We obtain multiple predictions using various numbers of shots in $D^j$ and perform majority voting to arrive at the final prediction. Details of this approach are shown in Algorithm 2 of the appendix.

## 3 Experiments

### 3.1 Experimental Settings

**Dataset** We conduct comprehensive experiments on the following four semantic parsing datasets:

- **Spider** (Yu et al., 2018) is a cross-domain semantic parsing dataset that contains complex Text-to-SQL problems. The data originates from 200 databases covering 138 different domains. We use the 7,000 training data as our pool of annotated examples.

- **Spider-Syn** (Gan et al., 2021a) replaced schema-related words in the questions of Spider examples with manually selected synonyms that reflect real-world question paraphrases to evaluate the robustness of systems.

- **Spider-DK** (Gan et al., 2021b) defined five types of domain knowledge and modified some Spider examples by adding domain knowledge to evaluate the cross-domain generalization capability of a given system.

- **Spider-Realistic** (Deng et al., 2021) removed explicit mentions of column names from Spider examples to reflect more realistic text-table alignment settings, and selected eight existing Text-to-SQL datasets for cross-domain evaluation.

**Model** We evaluate different ICL strategies with Codex (Chen et al., 2021), a GPT-3 variant that was finetuned on code data on the web and has demonstrated state-of-the-art performance as the time of writing (Ni et al., 2023). Specifically, we use the `code-davinci-002` engine and present the results of systems with prompts ranging from 1 to 10-shot. Additionally, we report the few-shot results utilizing the ChatGPT (`gpt-3.5-turbo`) model. However, due to its maximum context length limitation of 4096, we only obtain results for systems provided with prompts ranging from 1 to 5-shot.[4]

**Evaluation Metric** We use *execution accuracy* as the evaluation metric for all experiments, which measures the percentage of system predictions leading to the gold execution result.

**Baselines** We compare the following prompting strategies for generating SQL queries in few-shot and zero-shot settings.

*Few-shot*

- *Random sampling (R)*: Select demonstration examples randomly from the pool.
- *Similarity sampling (S)*
- *Diversity sampling (D)*: Select diverse examples from $k$-Means clusters of the pool.
- *Similarity-Diversity sampling (SD)*: Select examples based on Algorithm 1.
- *SD + schema augmentation (SA)*: Enhance instructions with schema knowledge (semantic augmentation or structure augmentation).

---

[4]Public API available at `https://openai.com/api/`.

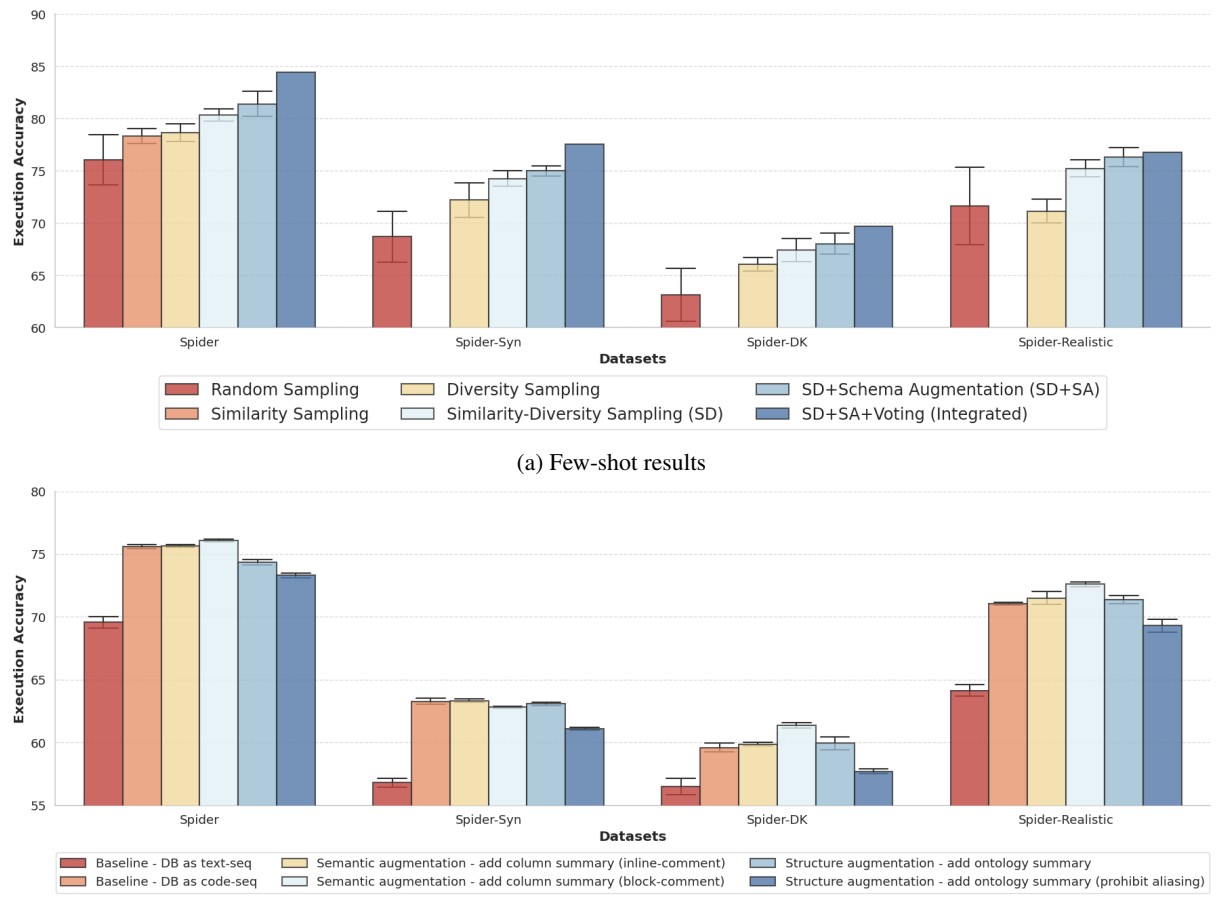

(a) Few-shot results

(b) Zero-shot results

Figure 1: Few-shot and zero-shot results of Codex for all datasets. In the few-shot setting, error bars indicate means and standard deviations over performances of systems provided with prompts ranging from 4-shot to 10-shot. To obtain the error bars for the random sampling approach, we conducted 3 independent runs using different random seeds. Schema augmentation utilized for the reported results in (a) is structure augmentation - add ontology summary. In the zero-shot setting, the error bars indicate means and standard deviations over 3 independent runs. Our results suggest that 1) using similarity and diversity objectives in the sampling process, 2) including schema representation in instructions, and 3) employing model voting with different shot outcomes both contribute to the improvement of ICL performance.

- *SD + SA + Voting*: Integrated strategy described in Algorithm 2.

*Zero-shot*

- *Baseline - DB as text-seq*: Standard prompt for Text-to-SQL task, where structured knowledge is linearized as text sequence.
- *Baseline - DB as code-seq*: Improve instructions by linearizing structured knowledge source as multiple SQL CREATE queries.
- *Baseline - DB as code-seq + SA*: Enhance instructions with schema knowledge.

## 3.2 Main Results

In this section, we present a comprehensive analysis of various prompting strategies, assessing their efficacy across multiple datasets. The evaluation of demonstration sampling strategies in a few-shot setting testing on `code-davinci-002` is illustrated in Figure 1a, and more few-shot results of `gpt-3.5-turbo` are shown in Figure 2. We compared different demonstration selection strategies, including random selection, $k$-nearest neighbors selection (similarity sampling)[5], $k$-means selection (diversity sampling), and our proposed approach, which combines both similarity and diversity. Moreover, we examined the impact of augmenting schema representation within the task instructions and assessed the performance of our

---
[5]Due to the deprecation of the Codex API in March 2023, similarity sampling experiments were only conducted on the Spider dataset.

integrated strategy. Our findings indicate that employing similarity and diversity objectives in the sampling process leads to better performance on average across all datasets. Furthermore, incorporating schema representation within the instructions enhances performance, and the implementation of voting of models with different shot results in a marked improvement in overall performance.

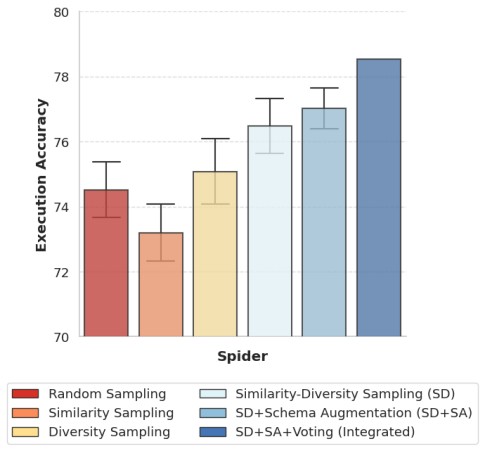

Figure 2: Few-shot results of `gpt-3.5-turbo` for Spider. Error bars indicate means and standard deviations over performances of systems provided with 1-shot to 5-shot prompts. Schema augmentation utilized for the reported results is semantic augmentation - add column summary as block-comment.

The efficacy of schema augmentation is further supported by experiments in a zero-shot setting, as illustrated in Figure 1b. We compared systems using different linearization methods for prompts: one that transforms the database into a text sequence, and another that uses multiple `CREATE` queries to represent the database. The latter method shows noticeable improvement in performance. We also contrasted two separate techniques for augmenting schema representation: one that adds semantic information to each column within each table, and another that incorporates entity-relationship knowledge into the schema. The results suggest that structural augmentation (adding ontology summary) brings a slight greater improvement in the few-shot setting for Codex (shown in Figure 5), while semantic augmentation (adding column summary as block comments) proves more beneficial in the zero-shot setting for Codex and also the few-shot setting for ChatGPT (`gpt-3.5-turbo`). We hypothesize that this difference may arise from the less descriptive nature of structural augmentation, which calls for more

demonstrations in order to effectively understand and utilize the provided information. In future study, we will explore better structural schema augmentation that aligns to the zero-shot setting.

## 4  Analysis

### 4.1  Prediction-Syntax based Retrieval

The existing method for selecting demonstrations relies on the semantic representations of the question and the database. We propose an alternative method specifically for code generation tasks, which focuses on the syntax of the solution code. We examined syntax coverage and syntax similarity of the prompts produced with different strategies. Syntax coverage is computed by counting the occurrence of syntactic elements (keywords, operators, and identifiers), and dividing it by the total number of all syntactic elements. Syntax similarity, on the other hand, is measured by the mean Euclidean distance between the discrete vector representation of the predicted SQL and vectors that represent the gold SQLs of the demonstrations selected. As indicated in Table 1 of the appendix, both of these metrics contribute to the quality of the examples selected. Furthermore, a simple summation of the two measurements suggests a correlation with the system's performance, as illustrated in Figure 6 of the appendix. We argue the efficacy of our strategy through the following rationale: (1) in cases where the pool of annotated examples is limited in diversity of the problem structures, certain test problems may lack similar examples available for retrieval; and (2) neither the semantic representation of the question/database nor the distance metric inherently support encapsulation and comparison of different problem structures, whereas SQL syntax provides direct measurement of the problem structures. Given these constraints, the optimal strategy is to select similar examples while ensuring the coverage of as many syntax demonstrations as feasible to mitigate potential failures in similarity-based retrieval.

### 4.2  Comparative Analysis of Retrieval Methods

We conducted an examination of various similarity-based retrieval methods and presented a comparative analysis of their performance in Figure 3. The primary variable in this investigation was the representation extracted for each example, with a focus on extracting and comparing the following

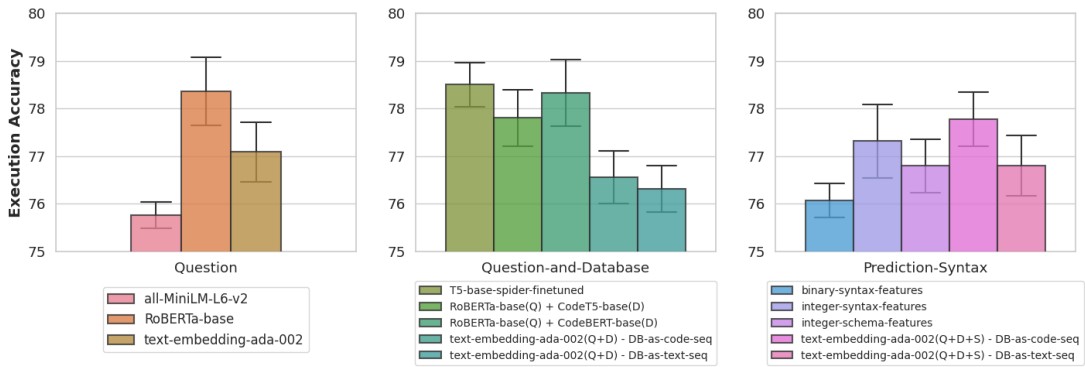

Figure 3: Comparison between various similarity based demonstration selection methods. Q indicates the embedding model employed to extract representation for the question; D stands for database, and S stands for SQL query.

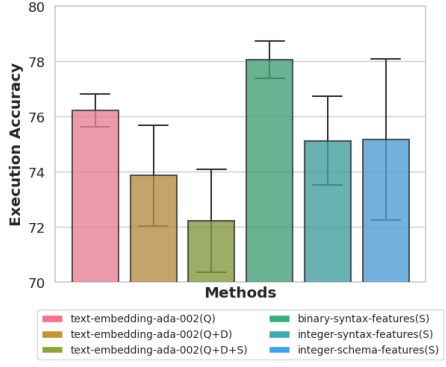

Figure 4: Comparison between various diversity based demonstration selection methods.

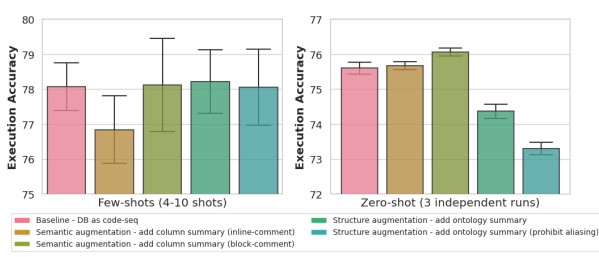

Figure 5: Comparison between various schema augmentations in few-shot and zero-shot settings.

embedding types: (1) question embeddings generated by Sentence-BERT (Reimers and Gurevych, 2019)[6] , RoBERTa-base (Liu et al., 2020), and OpenAI's text-embedding-ada-002; (2) combined question and database embeddings obtained by (i) employing a single model (i.e., T5-base (Raffel et al., 2020) finetuned on the Spider training split and text-embedding-ada-002) with the database linearized as a text sequence or CREATE queries, and (ii) utilizing separate models, specifically RoBERTa-base for encoding questions and

CodeT5-base (Wang et al., 2021) or CodeBERT-base (Feng et al., 2020) for encoding databases; (3) syntactic embeddings of predicted SQL, generated by either binary coding to indicate the presence of SQL syntactic elements or by quantifying their occurrences; and finally, (4) embeddings that encode questions, databases and predicted SQL using text-embedding-ada-002.

The following conclusions can be drawn about the similarity-based retrieval methods for Text-to-SQL task: (1) questions alone effectively represent distinct examples for retrieval purposes; (2) RoBERTa-base provides better embeddings for comparisons relative to text-embedding-ada-002; (3) it is feasible to employ models that have not been fine-tuned on Text-to-SQL examples for similarity-based retrieval, while still achieving comparable performance to fine-tuned models; (4) the linearization of databases as SQL queries facilitates the extraction of enhanced embeddings.

Additionally, we conducted a comparison between multiple embeddings utilized for diversity-based demonstration selection, encompassing embeddings that encode the semantics of questions, databases and predicted SQL, as well as embeddings that capture the syntactic features of predicted SQL. As depicted in Figure 4, the syntactic embeddings of predicted SQL serve as the most effective basis for contrasting different examples for diversity-based retrieval purposes.

### 4.3 Schema Augmentation

Figure 5 presents the outcomes of various schema augmentations applied to the instruction. It is observed that improvement is not apparent in the few-shot setting; however, in the zero-shot setting, the

---

[6]HuggingFace model name: all-MiniLM-L6-V2

semantic augmentation incorporating descriptions of all table columns proves to be beneficial.

## 4.4 Effectiveness Analysis

In order to determine the problem types that benefit most or least from our proposed methods, we also evaluate the performance of different models across various problem categories within the Spider dataset. As indicated in Figure 7 of the appendix, our similarity-diversity strategy proves beneficial for most problem types, with the exception of the medium split, which includes the most diverse problems. This is the case where similarity-based retrieval fails and syntax coverage becomes more crucial. Furthermore, we observe that augmenting schema semantics is more effective for the easy and medium splits (albeit with high variance), while augmenting schema structure is more effective for more complex problems. This observation leads us to hypothesize that challenging problems necessitate addressing a higher number of tables, thus requiring a more comprehensive understanding of the entire database structure. Lastly, the integrated approach is effective across all examples, offering increased benefits especially for those difficult problems.

## 4.5 Preliminary Models

To assess the impact of the choice of preliminary model used to generate the draft SQL on our approach, we conducted tests involving our methods for preliminary models with varying performance levels. Figure 8 of the appendix reveals that the preliminary models have a relatively minor effect on the performance of the similarity-diversity or integrated approaches, exhibiting gradual improvements as better preliminary models are utilized.

## 5 Related Work

Existing literature indicates the ability of large language models to adapt to new tasks at inference time by learning from a few example demonstrations (Brown et al., 2020; Radford et al., 2019). This new capability has been referred to as *in-context learning*. In this paper, we expand on previous works that investigate the optimal representations for prompt inputs.

### 5.1 Prompt Organization

Prompt organization investigates the task of selecting and organizing in-context examples, a critical aspect of enhancing model performance. Several studies (Sorensen et al., 2022; Gonen et al., 2022; Wu et al., 2022; Hu et al., 2022; Lu et al., 2022) have proposed metrics to measure the suitability of examples with respect to the target objective and to determine the optimal ordering of them. Liu et al. (2022) suggest selecting examples that are semantically similar to the test example by employing a $k$-NN approach in the embedding space. Rubin et al. (2022) train a prompt retriever based on contrastive learning, wherein examples are classified as either positive or negative if they are ranked among the top-$k$ or bottom-$k$ probabilities of a language model generating the target output, conditioned on the retrieved example and the input. Zhang et al. (2022) suggests to actively select demonstrations using Q-Learning. Su et al. (2023) introduces the Vote-$k$ approach to selectively annotate diverse and representative examples for pool construction, then retrieve based on the similarity. In contrast, our approach retrieve a diverse set of examples given a pre-established pool. As the authors demonstrate that having a diverse and representative pool is important for the success of ICL, we posit that a similar characteristic is equally important when composing the prompt, as this approach increases the likelihood of including various syntactical usages or similar problem structures within the prompt.

### 5.2 Prompt Formatting

Prompt engineering is concerned with investigating the impact of prompt structure on downstream task performance. For tasks that involve multi-step reasoning and higher complexity, *Chain-of-thought* prompting has been developed (Wei et al., 2023; Kojima et al., 2023). This approach involves laying out the generation process over multiple steps and using the model's own intermediate process as input. Wang et al. (2023) proposes to sample multiple different chain-of-thoughts then selects the most consistent answer through marginalization of all possible reasoning paths. Press et al. (2023) suggests that prompting LLMs to ask follow-up questions is an effective way to construct the chain-of-thoughts process. Zhou et al. (2023) proposes an automatic approach to identify the optimal prompt by searching over a pool of model generated instructions, assigning scores to them, and selecting the prompt with the highest score.

# 6 Conclusions

In this study, we investigated various prompt design approaches for semantic parsing tasks in the Text-to-SQL domain. We proposed an approach that leverages an example's SQL syntactic structure for demonstration examples selection, emphasising both diversity and similarity as the sampling objectives. Additionally, We found that LLMs gain benefits from database-related knowledge augmentations. Future research can build upon our findings to examine the transferability of our approach to other domains. Through ongoing improvement of LLMs' capabilities in semantic parsing, we aim to contribute to the development of QA systems that are more accurate, robust and comprehensible.

## Limitations

One of the main limitations of this study is the reproducibility problem. The experiments presented in this paper relied on the use of OpenAI APIs, which were available at the time of our research but have since been or will be deprecated. This means that the results of our experiments cannot be replicated using the same APIs, which hinders the reproducibility of our findings. To address this limitation, we will focus on providing experiments results that are based on open-sourced LLMs (Touvron et al., 2023; Taori et al., 2023; Chiang et al., 2023) for greater transparency and reproducibility.

Another limitation is that it is not clear how our approach will benefit LLMs given smaller or more constrained pools of annotated examples. Although we postulate that our approach offers the advantage of providing a prompt with maximal coverage of similar problem structures when identically structured problems cannot be found in the pool, we could not substantiate this due to our limited budget and access to the OpenAI APIs.

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

# Appendix

---

**Algorithm 1:** Similarity-Diversity Demonstration Selection

---

**Input:** Set of annotated examples $A$, test examples $T$, # demonstrations $k$, categorization $\{\alpha, \beta, ...\}$

**Result:** Set of prompts $P$, where $P_i$ is the prompt for test example $T_i$

```
/* Split A into disjoint partitions A^{α,β,···}                              */
```
**for** $A_i$ *in annotated set* $A$ **do**
  $c_i = $ `get_category`$(A_i.SQL)$;
  $A^{c_i}$`.append`$(A_i)$;
  $V_i = $ `get_syntax_vectors`$(A_i.SQL)$;
**end**
```
/* Prepare demonstrations D^j for each partition A^j                        */
```
**for** *partition* $A^j$ *in* $A^{\alpha, \beta, ···}$ **do**
  $M = k$-`Means_clustering`$(V^j, k)$;
```
  /* V^j is set of discrete vectors for examples in A^j, M has k centroids
     μ_1, ..., μ_k                                                           */
```
  **for** $\mu_i$ *in* $M$ **do**
    $D^j$`.append`(`get_nearest`$(A, \mu_i)$);
  **end**
**end**
```
/* Build test prompts                                                        */
```
**for** $T_i$ *in test set* $T$ **do**
  $T_i.SQL = $ `initial_predictor`$(T_i)$;
  $c_i = $ `get_category`$(T_i.SQL)$;
  $P_i = $ `build_prompt`$(D^{c_i}, T_i)$;
**end**
**return** $P$

---

**Algorithm 2:** Integrated Strategy

---

**Input:** Set of annotated examples $A$, test examples $T$, # demonstrations $k$, categorization $\{\alpha, \beta, ...\}$, and from Algorithm 1: disjoint partitions $\{A^\alpha, A^\beta, ...\}$ and corresponding demonstrations $\{D^\alpha, D^\beta, ...\}$

**Result:** Set of SQL predictions $SP$, where $SP_i$ is the final prediction for test example $T_i$

**for** $T_i$ *in test set* $T$ **do**

    $T_i.SQL = \texttt{initial\_predictor}(T_i)$;

    $c_i = \texttt{get\_category}(T_i.SQL)$;

    **for** $n = 4$ **to** $k$ **do**

        $P_i^n = \texttt{build\_prompt}(D^{c_i}[:n], T_i)$;

        $P_i^{n^*} = \texttt{augment\_schema}(P_i^n)$;

        $SP_i^n = \texttt{Model}(P_i^{n^*})$;

        $ER_i^n = \texttt{DBMS}(SP_i^n)$;

    **end**

    $ER_i^* = \texttt{Remove\_Exec\_Errors}(ER_i)$;

    $SP_i = \texttt{Majority\_Vote}(ER_i^*)$;

**end**

**return** $SP$

---

| | Coverage | Similarity | Execution Accuracy |
|---|---|---|---|
| **Random** | 0.38 | 0.24 | 76.03 |
| **Similarity** | 0.35 | 0.30 | 78.33 |
| **Diversity** | 0.43 | 0.23 | 78.64 |
| **Similarity-Diversity** | 0.50 | 0.26 | 80.32 |

Table 1: Average syntax coverage and similarity measures of the prompt for different demonstration selection strategies and the corresponding execution accuracies.

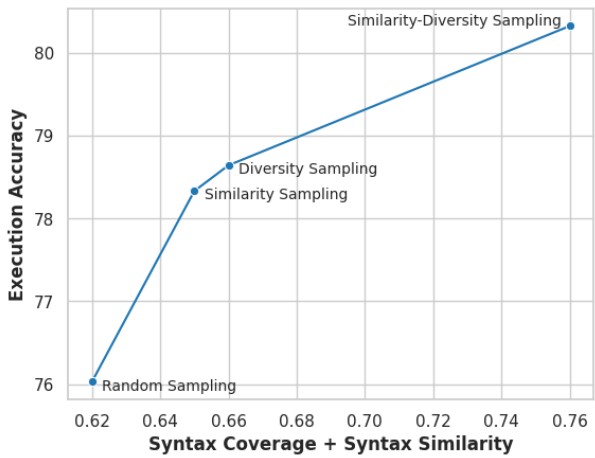

Figure 6: Correlation between syntax coverage and similarity measures of prompts and execution accuracy.

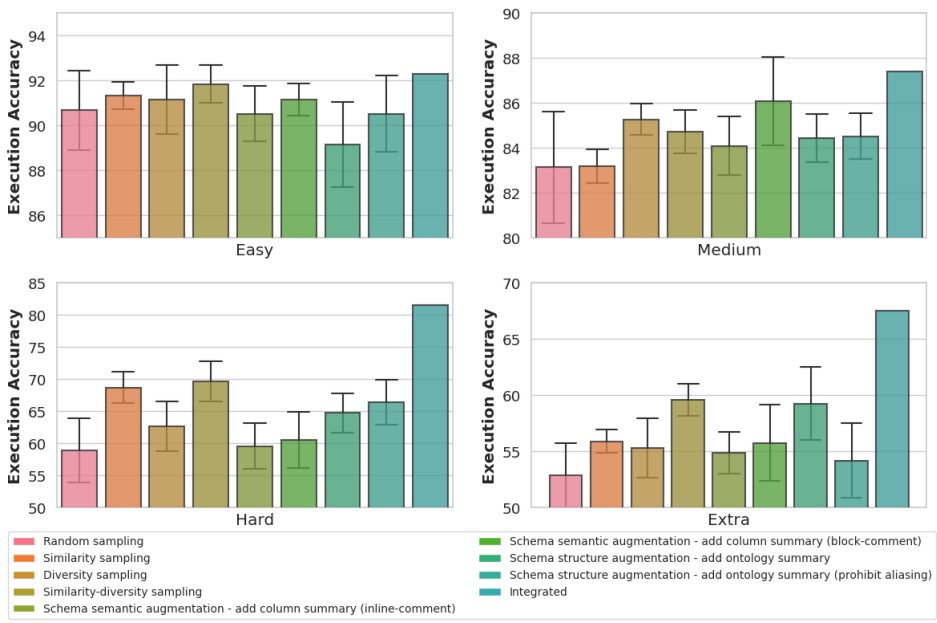

Figure 7: Effects of various prompting strategies on Text-to-SQL problems of different difficulty levels.

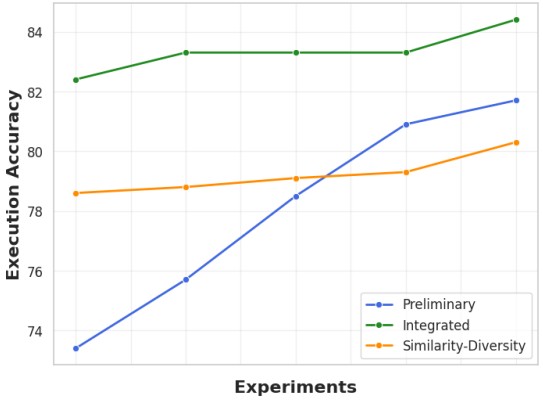

Figure 8: Effects of preliminary model on proposed strategies.

```
1   Given the following database schema:
2   gymnast : gymnast_id, floor_exercise_points, pommel_horse_points, rings_points,
            vault_points, parallel_bars_points, horizontal_bar_points, total_points |
        people : people_id, name, age, height, hometown
3
4   Answer the following: Return the total points of the gymnast with the lowest
        age.
5
6   select t1.total_points from gymnast as t1 join people as t2 on t1.gymnast_id =
        t2.people_id order by t2.age asc limit 1
```

Listing 1: Baseline prompt with text representation of the database.

```
1   /* Given the following database schema: */
2   CREATE TABLE IF NOT EXISTS "gymnast" (
3       "Gymnast_ID" int,
4       "Floor_Exercise_Points" real,
5       "Pommel_Horse_Points" real,
6       "Rings_Points" real,
7       "Vault_Points" real,
8       "Parallel_Bars_Points" real,
9       "Horizontal_Bar_Points" real,
10      "Total_Points" real,
11      PRIMARY KEY ("Gymnast_ID"),
12      FOREIGN KEY ("Gymnast_ID") REFERENCES "people"("People_ID")
13      );
14  CREATE TABLE IF NOT EXISTS "people" (
15      "People_ID" int,
16      "Name" text,
17      "Age" real,
18      "Height" real,
19      "Hometown" text,
20      PRIMARY KEY ("People_ID")
21  );
22
23  /* Answer the following: Return the total points of the gymnast with the lowest
        age. */
24
25  select t1.total_points from gymnast as t1 join people as t2 on t1.gymnast_id =
        t2.people_id order by t2.age asc limit 1
```

Listing 2: Baseline prompt with code representation of the database.

```
1   /* Given the following database schema: */
2   CREATE TABLE IF NOT EXISTS "department" (
3       "Department_ID" int, -- a unique identifier for a department
4       "Name" text, -- the name of the department
5       "Creation" text, -- the date the department was created
6       "Ranking" int, -- the ranking of the department within the organization
7       "Budget_in_Billions" real, -- the department's budget in billions of
            dollars
8       "Num_Employees" real, -- the number of employees in the department
9       PRIMARY KEY ("Department_ID")
10  );
11  CREATE TABLE IF NOT EXISTS "head" (
12      "head_ID" int, -- a unique identifier for the head of a department
13      "name" text, -- the name of the head of the department
14      "born_state" text, -- the state where the head of the department was born
15      "age" real, -- the age of the head of the department
16      PRIMARY KEY ("head_ID")
17  );
18  CREATE TABLE IF NOT EXISTS "management" (
19      "department_ID" int, -- the unique identifier for the department being
            managed
20      "head_ID" int, -- the unique identifier for the head of the department
21      "temporary_acting" text, -- whether the head of the department is serving
            in a temporary or acting capacity
22      PRIMARY KEY ("Department_ID", "head_ID")
23      FOREIGN KEY ("Department_ID") REFERENCES `department`("Department_ID")
24      FOREIGN KEY ("head_ID") REFERENCES `head`("head_ID")
25  );
26
27  /* Answer the following: What are the distinct creation years of the
        departments managed by a secretary born in state 'Alabama'? */
28
29  select distinct t1.creation from department as t1 join management as t2 on t1.
        department_id = t2.department_id join head as t3 on t2.head_id = t3.head_id
         where t3.born_state = 'Alabama'
```

Listing 3: Prompt with semantic augmentation of the schema as inline comment.

```
1   /* Given the following database schema: */
2   CREATE TABLE IF NOT EXISTS "department" (
3       "Department_ID" int,
4       "Name" text,
5       "Creation" text,
6       "Ranking" int,
7       "Budget_in_Billions" real,
8       "Num_Employees" real,
9       PRIMARY KEY ("Department_ID")
10  );
11  CREATE TABLE IF NOT EXISTS "head" (
12      "head_ID" int,
13      "name" text,
14      "born_state" text,
15      "age" real,
16      PRIMARY KEY ("head_ID")
17  );
18  CREATE TABLE IF NOT EXISTS "management" (
19      "department_ID" int,
20      "head_ID" int,
21      "temporary_acting" text,
22      PRIMARY KEY ("Department_ID","head_ID"),
23      FOREIGN KEY ("Department_ID") REFERENCES `department`("Department_ID"),
24      FOREIGN KEY ("head_ID") REFERENCES `head`("head_ID")
25  );
26
27  /* Table column descriptions:
28  {'department': {'Department_ID': 'a unique identifier for a department', 'Name
        ': 'the name of the department', 'Creation': 'the date the department was
        created', 'Ranking': 'the ranking of the department within the organization
        ', 'Budget_in_Billions': "the department's budget in billions of dollars",
        'Num_Employees': 'the number of employees in the department'}, 'head': {'
        head_ID': 'a unique identifier for the head of a department', 'name': 'the
        name of the head of the department', 'born_state': 'the state where the
        head of the department was born', 'age': 'the age of the head of the
        department'}, 'management': {'department_ID': 'the unique identifier for
        the department being managed', 'head_ID': 'the unique identifier for the
        head of the department', 'temporary_acting': 'whether the head of the
        department is serving in a temporary or acting capacity'}} */
29  /* Answer the following: What are the distinct creation years of the
        departments managed by a secretary born in state 'Alabama'? */
30
31  select distinct t1.creation from department as t1 join management as t2 on t1.
        department_id = t2.department_id join head as t3 on t2.head_id = t3.head_id
         where t3.born_state = 'Alabama'
```

Listing 4: Prompt with semantic augmentation of the schema as block comment.

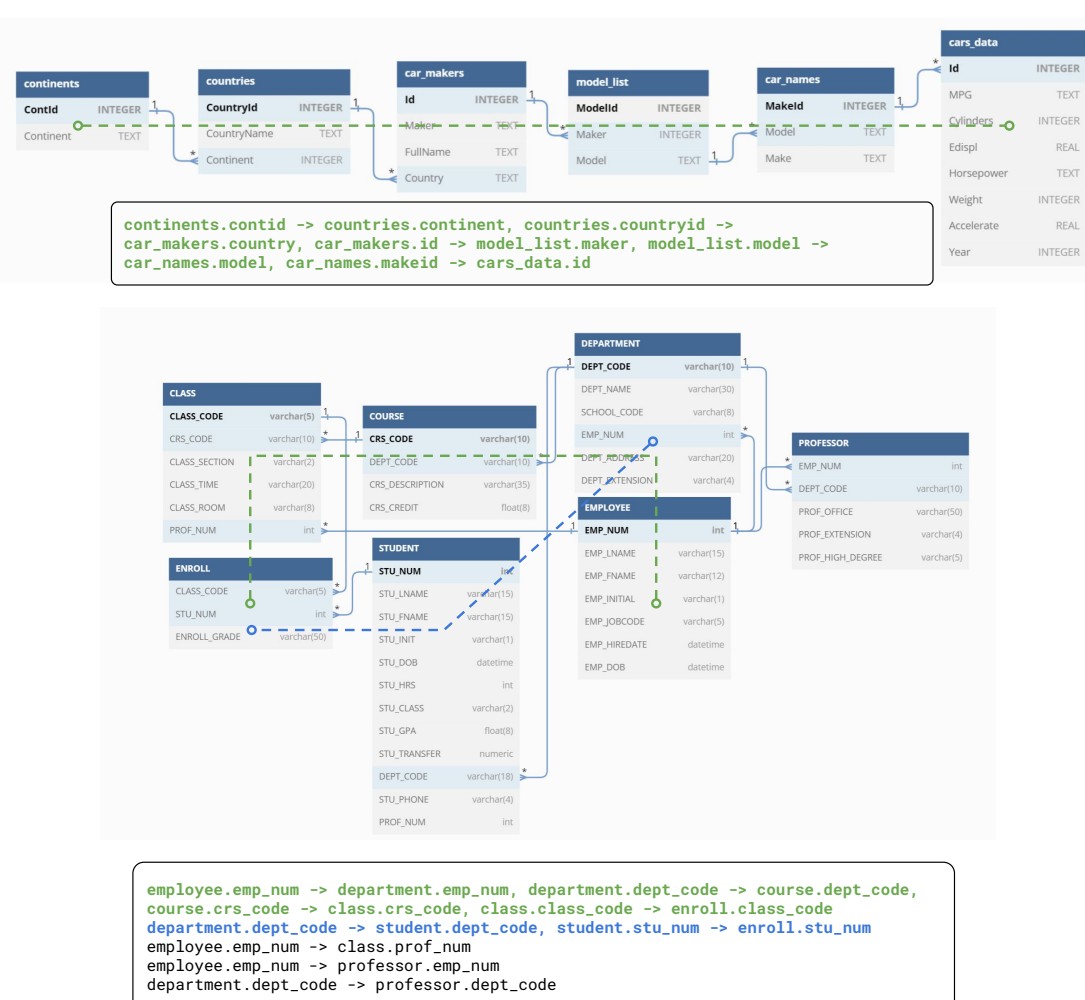

Figure 9: Examples of schema structure representation construction.

```sql
/* Given the following database schema: */
CREATE TABLE IF NOT EXISTS "continents" (
    "ContId" INTEGER PRIMARY KEY,
    "Continent" TEXT
);
CREATE TABLE IF NOT EXISTS "countries" (
    "CountryId" INTEGER PRIMARY KEY,
    "CountryName" TEXT,
    "Continent" INTEGER,
    FOREIGN KEY (Continent) REFERENCES continents(ContId)
);
CREATE TABLE IF NOT EXISTS "car_makers" (
    "Id" INTEGER PRIMARY KEY,
    "Maker" TEXT,
    "FullName" TEXT,
    "Country" TEXT,
    FOREIGN KEY (Country) REFERENCES countries(CountryId)
);
CREATE TABLE IF NOT EXISTS "model_list" (
    "ModelId" INTEGER PRIMARY KEY,
    "Maker" INTEGER,
    "Model" TEXT UNIQUE,
    FOREIGN KEY (Maker) REFERENCES car_makers (Id)

);
CREATE TABLE IF NOT EXISTS "car_names" (
    "MakeId" INTEGER PRIMARY KEY,
    "Model" TEXT,
    "Make" TEXT,
    FOREIGN KEY (Model) REFERENCES model_list (Model)
);
CREATE TABLE IF NOT EXISTS "cars_data" (
    "Id" INTEGER PRIMARY KEY,
    "MPG" TEXT,
    "Cylinders" INTEGER,
    "Edispl" REAL,
    "Horsepower" TEXT,
    "Weight" INTEGER,
    "Accelerate" REAL,
    "Year" INTEGER,
    FOREIGN KEY (Id) REFERENCES car_names (MakeId)
);

/*
Database ontology:
continents.contid -> countries.continent, countries.countryid -> car_makers.
    country, car_makers.id -> model_list.maker, model_list.model -> car_names.
    model, car_names.makeid -> cars_data.id
*/
/* Answer the following: How many continents are there? */

select count(*) from continents;
```

Listing 5: Prompt with structure augmentation of the schema.

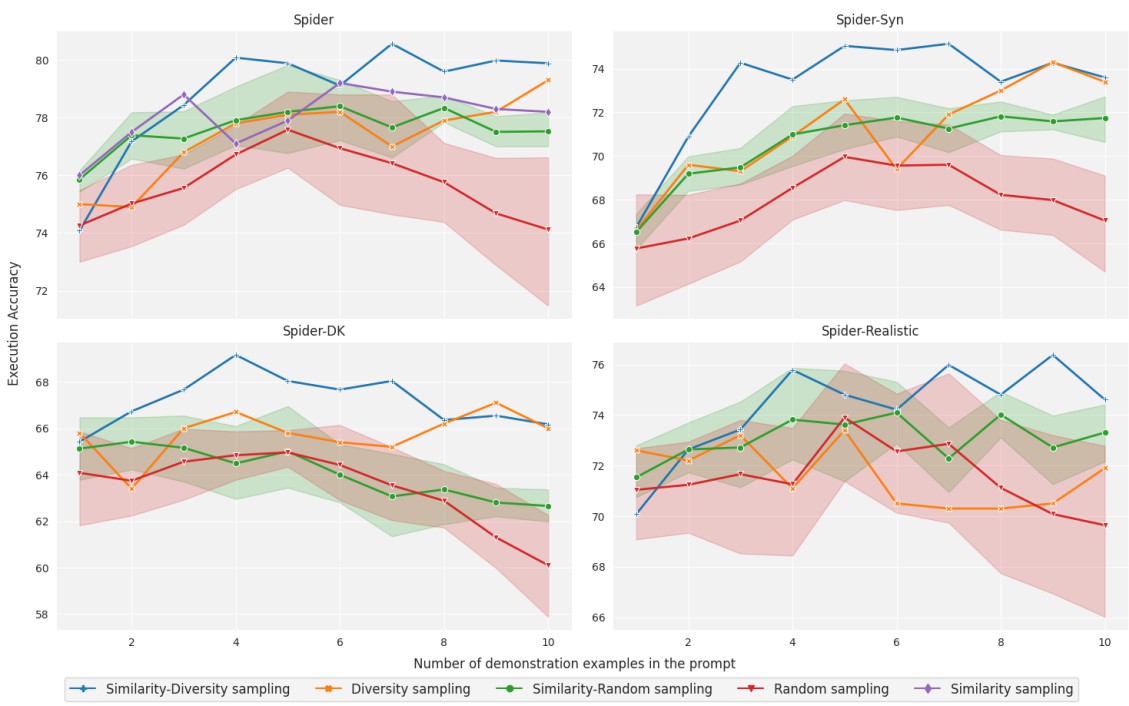

Figure 10: Few-shot results for comparing different sampling strategies with different number of demonstration examples selected for the prompt.

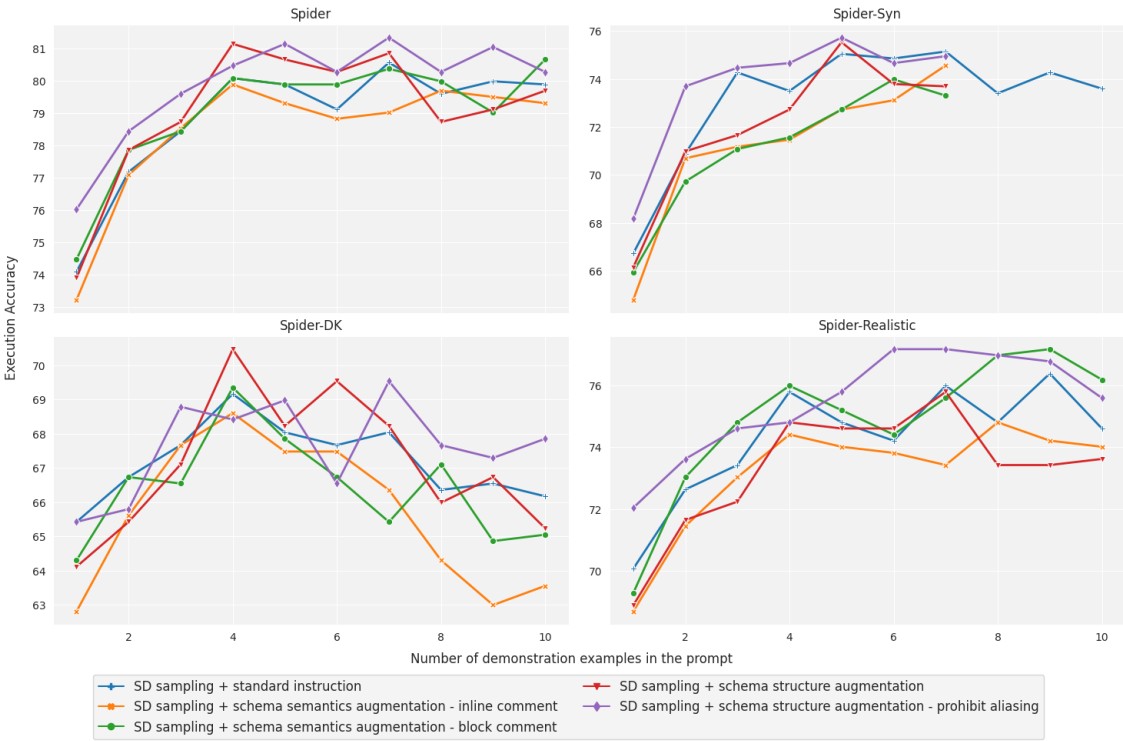

Figure 11: Few-shot results for comparing different schema representation augmentation methods with different number of demonstration examples selected for the prompt.

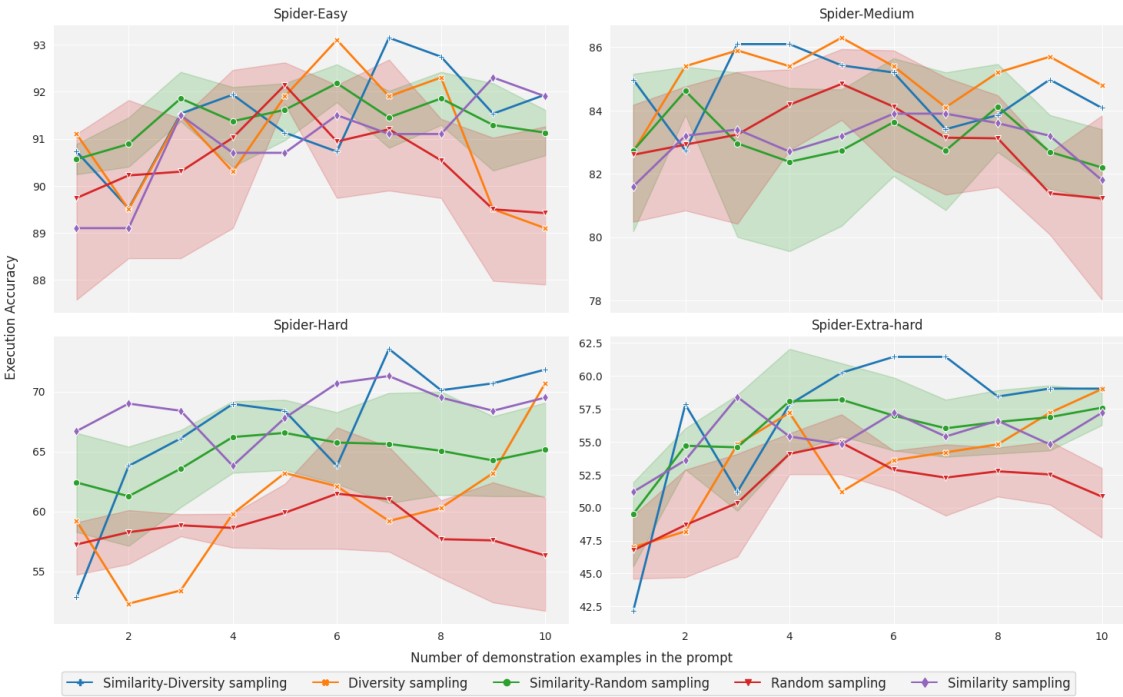

Figure 12: Few-shot results for comparing different sampling strategies on Text-to-SQL problems of different difficulty levels, with different number of demonstration examples selected for the prompt.