# OpenReview forum: "Enhancing Text-to-SQL Capabilities of Large Language Models: A Study on Prompt Design Strategies"
_EMNLP/2023/Conference — EMNLP 2023 Findings_

### Official Review · Reviewer_rZ1M · 2023-07-26

**Typos Grammar Style And Presentation Improvements:** N/A
**Soundness:** 4

**Excitement:**

4: Strong: This paper deepens the understanding of some phenomenon or lowers the barriers to an existing research direction.

**Missing References:**

The best fine-tuned system was [1], which was released more than 3 months ahead of the EMNLP2023 deadline.
[1] Li, Haoyang, et al. "Resdsql: Decoupling schema linking and skeleton parsing for text-to-sql." Proceedings of the AAAI Conference on Artificial Intelligence. Vol. 37. No. 11. 2023.

**Paper Topic And Main Contributions:**

The paper explores prompt design strategies for text-to-SQL with in-context learning. The authors propose to retrieve demonstrations with SQL syntactic structure. Given a test instance, the proposed method first generates a preliminary SQL query and uses the preliminary SQL query to retrieve demonstration examples. For the retrieval strategy, the authors propose to consider both the diversity of demonstrations and the similarity between demonstrations and the test instance. The paper also studies how to represent schema-related knowledge in prompt designs in a zero-shot scenario.
By combining schema-related knowledge and the demonstration retrieval method, the proposed method outperforms the previous state-of-the-art in-context learning methods and fine-tuned methods. The findings in the paper will benefit future study text-to-SQL.



**Questions For The Authors:**

1. How does the Codex perform when retrieving demonstrations with the oracle SQL query of the test instance?
2. How many experiments are run to achieve the majority voting for one test instance?

**Reasons To Accept:**

1. The paper compares various demonstration retrieval strategies for text-to-SQL with in-context learning and reveals that the most preferred demonstrations are to consider both similarity and diversity.
2. The paper also studies how to represent schema-related knowledge in the zero-shot prompt designs.
3. The experiments on multiple text-to-SQL datasets demonstrate the generalization of the findings.
4. The paper is well-written and easy to follow.

**Reasons To Reject:**

1. The authors should be more careful when saying "we propose". (it does not sound like a problem to me once some editing is done).
(1) In Line 224, the authors say "We propose ... code sequence ... CREATE query". The CREATE query for database schema was proposed by [1]. The authors should be aware of this work as it is cited in the introduction section.
(2) Retrieve diverse demonstrations for semantic parsing has been proposed in this work [2], although it is not focusing on text-to-SQL.

[1] Rajkumar, Nitarshan, Raymond Li, and Dzmitry Bahdanau. "Evaluating the text-to-sql capabilities of large language models." arXiv preprint arXiv:2204.00498 (2022).
[2] Levy, Itay, Ben Bogin, and Jonathan Berant. "Diverse demonstrations improve in-context compositional generalization." arXiv preprint arXiv:2212.06800 (2022).

**Reproducibility:**

3: Could reproduce the results with some difficulty. The settings of parameters are underspecified or subjectively determined; the training/evaluation data are not widely available.

**Reviewer Confidence:**

5: Positive that my evaluation is correct. I read the paper very carefully and I am very familiar with related work.

---

> ### Author Rebuttal · Authors · 2023-08-29
>
> Thank you for your constructive feedback and valuable suggestions! They have been incredibly useful, and we will integrate them into the updated manuscript. In the following, we would like to address your comments and questions.
>
> > The authors should be more careful when saying "we propose". (it does not sound like a problem to me once some editing is done). (1) In Line 224, the authors say "We propose ... code sequence ... CREATE query". The CREATE query for database schema was proposed by [1]. The authors should be aware of this work as it is cited in the introduction section. (2) Retrieve diverse demonstrations for semantic parsing has been proposed in this work [2], although it is not focusing on text-to-SQL.
>
> We would like to thank the reviewer for pointing this out and we agree that we should change the phrasing “we propose” and mention Rajkumar et al., 2022 regarding CREATE query approach. We also appreciate the reviewer for the second reference which we overlooked, and we will add it to the revised manuscript.
>
> > How does the Codex perform when retrieving demonstrations with the oracle SQL query of the test instance?
>
> We have not conducted experiments of Codex retrieving demonstrations with the test-set oracle SQL queries directly, and will conduct them if we regain access to Codex. But with draft SQL, we are able to obtain 74% accuracy on difficulty-level prediction, indicating that for 74% of the test instances, we retrieved the same demonstrations as those retrieved using oracle SQL and therefore leading to the same results.
>
> > How many experiments are run to achieve the majority voting for one test instance?
>
> We obtain 7 experiment results (4-10 shots) to achieve the majority voting for one test instance. We will add this clarification in the revised manuscript.
>
> > The best fine-tuned system was [1], which was released more than 3 months ahead of the EMNLP2023 deadline. [1] Li, Haoyang, et al. "Resdsql: Decoupling schema linking and skeleton parsing for text-to-sql." Proceedings of the AAAI Conference on Artificial Intelligence. Vol. 37. No. 11. 2023.
>
> We appreciate the reviewer for pointing out the missing reference, and we will add it to the revised manuscript.

---

### Official Review · Reviewer_FoCC · 2023-08-04

**Typos Grammar Style And Presentation Improvements:** 1. The title can be further refined. …
**Soundness:** 3

**Excitement:**

3: Ambivalent: It has merits (e.g., it reports state-of-the-art results, the idea is nice), but there are key weaknesses (e.g., it describes incremental work), and it can significantly benefit from another round of revision. However, I won't object to accepting it if my co-reviewers champion it.

**Paper Topic And Main Contributions:**

This submission presents a novel approach to enhance the Text-To-SQL task by incorporating two key components: demonstration-selection and schema augmentation. It conducts experiments on semantic parsing datasets to prove the effectiveness of proposed method.

**Questions For The Authors:**

"Code" method seems like evaluated only as a baseline setting in zero-shot. What's the point to include this baseline?

**Reasons To Accept:**

1. The proposed demonstration-selection approach, which takes into account syntactic structure, diversity, and similarity, shows promise and can be of inspiring. It has the potential to provide valuable insights and inspiration for further advancements in the area of LLM based Text-To-SQL tasks.
2. The proposed method demonstrates notable improvements on semantic parsing datasets.

**Reasons To Reject:**

1. The comparison is conducted against a customized baseline, making it challenging to directly align the results with previous work, such as Liu et al., 2023. This limitation also applies when attempting to validate the potential of the proposed demonstration-selection method within a few-shot setting. While I believe that previous in-context demonstration-selection approaches might not be well-suited for this Text-To-SQL task, it is essential to conduct fair comparisons to substantiate this claim with concrete evidence.
2. The claim that syntactic based retrieval is better than other methods lacks of support.
3. Without refinement, current way of organizing the paper makes me difficult to follow the exact method being proposed.


**Reproducibility:**

4: Could mostly reproduce the results, but there may be some variation because of sample variance or minor variations in their interpretation of the protocol or method.

**Reviewer Confidence:**

4: Quite sure. I tried to check the important points carefully. It's unlikely, though conceivable, that I missed something that should affect my ratings.

---

> ### Author Rebuttal · Authors · 2023-08-29
>
> Thank you for your constructive comments and suggestions, and they are exceedingly helpful for us to improve our paper! We have carefully incorporated them in the revised paper. Below, we address each of your concerns in detail.
>
> > The comparison is conducted against a customized baseline, making it challenging to directly align the results with previous work, such as Liu et al., 2023. This limitation also applies when attempting to validate the potential of the proposed demonstration-selection method within a few-shot setting. While I believe that previous in-context demonstration-selection approaches might not be well-suited for this Text-To-SQL task, it is essential to conduct fair comparisons to substantiate this claim with concrete evidence.
>
> We fully agree that aligning our results with previous work for fair comparisons is crucial for substantiating our claims. It's worth noting, however, that the resource constraints influenced the extent of our experimentation. While we were able to conduct few-shot evaluations for both Codex and ChatGPT, we presented zero-shot results for Codex only due to budget limitations.  Nonetheless, from Codex zero-shot results, we can already see that the baseline (DB as text seq) results on all the Spider variant datasets achieves similar performance compared to Liu et al's ChatGPT results (comparing EX scores) [1], and our proposed ontology-related augmentation can further improve them. Rajkumar et al., 2022 [2] also reports Codex results in zero-shot setting, similar to ours on the Spider dataset. These seem to be the only related prior work that study prompting with Codex/ChatGPT and reported results on Spider or variant datasets at the time of writing. Although Ni et al. [3] and Chen et al. [4] conducted few-shot experiments on Spider, their work diverges from ours by focusing on verification and self-debugging processes, which is an orthogonal investigation and not related to our claims.
> We plan to include these broader comparisons in our revised paper to provide a comprehensive understanding of our contribution. If the reviewer is aware of additional work that would require a comparison, we're open to including those as well.
>
> [1]: Aiwei Liu, Xuming Hu, Lijie Wen, and Philip S. Yu. 2023. A comprehensive evaluation of chatgpt’s zero-shot text-to-sql capability. arXiv preprint arXiv: 2303.13547
>
> [2]: Nitarshan Rajkumar, Raymond Li, and Dzmitry Bahdanau. 2022. Evaluating the text-to-sql capabilities of large language models. arXiv preprint arXiv: 2204.00498
>
> [3]: Ansong Ni, Srini Iyer, Dragomir Radev, Ves Stoyanov, Wen-tau Yih, Sida I. Wang, Xi Victoria Lin. 2023. LEVER: Learning to Verify Language-to-Code Generation with Execution. arXiv preprint arXiv: 2302.08468
>
> [4]: Xinyun Chen, Maxwell Lin, Nathanael Schärli, Denny Zhou. 2023. Teaching Large Language Models to Self-Debug. arXiv preprint arXiv: 2304.05128
>
> > The claim that syntactic based retrieval is better than other methods lacks of support.
> >
> > Without refinement, current way of organizing the paper makes me difficult to follow the exact method being proposed.
>
> It would be greatly helpful if the reviewer could specify which aspect of the claim that they find lacking in support, so that we can address it more effectively in the revision.
> Regarding paper organization, we proposed two prompt augmentations: 1. When selecting demonstrations, code syntax is a better source to extract representations, and introducing diversity in addition to the previous similarity-based retrieval approach helps; (Section 2.1) 2. adding ontology-related context of the database to the prompt helps. (Section 2.2) In addition, we propose a procedure to synergize these two augmentations. (Section 2.3) We will arrange these claims more clearly in the revision.
>
> > "Code" method seems like evaluated only as a baseline setting in zero-shot. What's the point to include this baseline?
>
> If the reviewer is referring to “DB as code-seq” in Figure 1(b), we linearize database as code-sequence for all other prompts (zero or few-shot), here we compare it with “DB as text-seq” in zero-shot setting to show that linearizing database schema to CREATE queries is indeed effective in improving the performance.
>
> > 1. The title can be further refined. Initially, it gives the impression of a technical report that compares various strategies without introducing any advanced approaches. Additionally, the paper proposes enhancements for both zero-shot and few-shot settings, and the mention of few-shot in the title might not be appropriate.
> >
> > 2. I believe there is room for improvement in conveying the message of the paper. While the current approach heavily relies on descriptions about the proposed method, incorporating simple examples would greatly enhance clarity. For instance, a example of the structured predictionas in Section 2.1 would be helpful for better understanding. Additionally, after the clustering process, including examples of the selected demonstrations will provide readers with a clearer insight into the proposed method.
> >
> > 3. There are certain details in experiments can be added, like how many testing samples in total, temperature setting for GPT, etc.
> >
> > 4. I have to switch back several times to align the terminology used in experiments like Schema Augmentation, Structure Augmentation and Semantic Augmentation with the one in methodology part. It will be good if highlight those terms can be highlithed.
>
> We appreciate the reviewer for bringing up these feedbacks! We will revise the paper to address these issues.

---

### Official Review · Reviewer_ZrTD · 2023-08-06

**Soundness:** 3

**Excitement:**

2: Mediocre: This paper makes marginal contributions (vs non-contemporaneous work), so I would rather not see it in the conference.

**Paper Topic And Main Contributions:**

This paper investigates techniques for improving performance of semantic parsing, specifically text-to-SQL, in in-context learning settings. It proposes (1) sampling techniques that take into account both similarity and diversity,  (2) different approaches to include schema information in prompts, and (3) ensemble approaches to aggregate results to tackle the sensitivity issue.

The evaluation is conducted on 5 different variants of the Spider dataset on Codex and ChatGPT.

**Questions For The Authors:**

1. Line 171, by "binary features" do you mean one-hot vectors? An example would help illustrate the idea here.

2. Line 197-199, you said "based on its category, retrieve candidate examples that belong to the relevant partition". How many candidate examples are retrieved here? Which part of Algorithm 1 does it correspond to?

3. Related, what is the rationale for using difficulty level as a proxy for (syntactic) similarity? I imagine questions in the same category level may be quite different syntactically. Have you investigated other similarity measures?

4. In the zero-shot setting, why does SD not work?

5. Line 496 and in Fig. 3 middle, why does RoBERTa have stronger performance than Ada, which, as a much larger model, should have better performance. Any insights?

6. What are the "preliminary models" in Sec. 4.5?

**Reasons To Accept:**

* Semantic parsing is an important NLP task with practical importance, as it allows neural models to interrogate real-world data. Thus, improving in-context learning performance is important for LLMs on this task.

**Reasons To Reject:**

* The technical contributions and novelty in this paper are limited. It empirically investigates different technical aspects that affect semantic parsing performance in ICL. Similar investigations have been conducted in many other tasks already, and there are not novel findings.

* The presentation of one of the main ideas in the paper, namely sample selection, is not clear. The diversity and similarity part (Algorithm 1 and associated description in Sec. 2.1) is confusing.

* The findings are not completely convincing or surprising.

**Reproducibility:**

3: Could reproduce the results with some difficulty. The settings of parameters are underspecified or subjectively determined; the training/evaluation data are not widely available.

**Reviewer Confidence:**

4: Quite sure. I tried to check the important points carefully. It's unlikely, though conceivable, that I missed something that should affect my ratings.

---

> ### Author Rebuttal · Authors · 2023-08-29
>
> Thank you for the time and effort you have dedicated to reviewing our work, we appreciate your constructive comments, which offer important perspectives for improving the quality of our paper! Below, we address each of your concerns in detail.
>
> > Line 171, by "binary features" do you mean one-hot vectors? An example would help illustrate the idea here.
>
> No, we don't mean one-hot vectors. Our binary feature vectors are more like “bag-of-syntactic-elements”, with each entry being 1 if a syntactic element is present in the SQL query (instead of the count), and 0 if not. For example, “select airportname from airports where airportcode = "AKO"” is converted to [1,1,1,1,0,0,...], where the first four entries correspond to the keywords “select”, “from”, “where” and “=”.
>
> > Line 197-199, you said "based on its category, retrieve candidate examples that belong to the relevant partition". How many candidate examples are retrieved here? Which part of Algorithm 1 does it correspond to?
>
> We adopt the difficulty-based categorization of SQL queries based on the implementation of Spider (Yu et al., 2018). We use Spider training split as the pool, which consists of 7,000 instances, from them we found 1,686 easy instances, 2,760 medium instances, 1,446 hard instances, 1,058 extra-hard instances, and 50 erroneous instances which we exclude from the pool. Other datasets use the same training split SQL queries as those in Spider, so the same statistics apply.
> Therefore, given a test instance, we use a preliminary model to generate a draft SQL, and then categorize it based on the aforementioned difficulty levels. For example, If the draft SQL is predicted to be easy, we retrieve the 1,686 easy instances as the candidates, from which we sample diverse demonstrations. This process corresponds to the first “for loop” in Algorithm 1.
>
> > Related, what is the rationale for using difficulty level as a proxy for (syntactic) similarity? I imagine questions in the same category level may be quite different syntactically. Have you investigated other similarity measures?
>
> We use the difficulty criteria defined and implemented in the Spider (Yu et al., 2018) as the proxy, because their categorization is developed strictly based on syntactic coverage and structure of a SQL query, ensuring that queries satisfying the same conditions are grouped into the same category. For instance, easy SQL queries should contain at most 1 keyword from WHERE, GROUP BY, ORDER BY, LIMIT, JOIN, OR, LIKE, HAVING and no keyword from EXCEPT, UNION, INTERSECT, NESTED and no more than 1 aggregation or column selection. We think this criteria already effectively groups SQL queries with similar syntactic structure and keyword coverage, and we did not explore semantic-based similarity because our main argument is that we can retrieve better Text-to-SQL demonstrations for prompting if we exploit the syntactic information of Text-to-SQL examples. We thank the reviewer for bringing up this question and we will provide more clarifications in the revised version of our paper.
>
> > In the zero-shot setting, why does SD not work?
>
> Not sure which result datapoint the reviewer is referring to. Similarity-Diversity sampling (SD) requires sampling demonstrations, so it only applies to few-shot settings. In zero-shot experiments, we only compare how adding different ontology-related context to the prompt would help.
>
> > Line 496 and in Fig. 3 middle, why does RoBERTa have stronger performance than Ada, which, as a much larger model, should have better performance. Any insights?
>
> We suspect that the poor performance of ada compared to RoBERTa-base, as shown in Figure 3(a, b), is due to its enhanced capability to retrieve semantically similar examples that are not necessarily syntactically similar, for example, it retrieves examples from the same database cued by similar lexicons, while these examples are likely to have different problem structures therefore would not be helpful demonstrations for solving the current test instance. We will assess the syntactic similarity of all demonstrations retrieved by RoBERTa and ada models to substantiate our hypothesis in the revised version of our paper.
>
> > What are the "preliminary models" in Sec. 4.5?
>
> Preliminary models are Text-to-SQL models that can be used to draft the SQL code. We can use any available Text-to-SQL baselines for drafting a SQL query. In Figure 8, we use baselines with different performance (blue lines) to see whether our method can be limited by applying a poor preliminary model. The results show that even with a poor draft (leftmost datapoint), the similarity-diversity approach and the integrated approach are still effective. We will provide more clarification about this in the revised version of our paper.
>
> > The technical contributions and novelty in this paper are limited. It empirically investigates different technical aspects that affect semantic parsing performance in ICL.
>
> We would like to re-emphasize our contributions. We identified two aspects that were ignored previously when designing prompts for Text-to-SQL tasks using LLMs, which can be extended to other code generation tasks: 1. If the output is code, then the syntactic information of the code is a better representation of the problem structure than the natural language query, therefore serves as a better basis for retrieving demonstrations. Most existing demonstration-retrieval methods, however, rely on query semantics. When retrieving demonstrations based on syntax, we could synergize similarity and the diversity objectives to retrieve examples that are more helpful to solve the test instance; 2. For database related tasks, it's also crucial to incorporate the database ontology (e.g., column type, foreign key constraints, etc.) into the prompt. Last but not least, we propose a procedure to synergize these two augmentations.
>
> > Similar investigations have been conducted in many other tasks already, and there are not novel findings.
>
> It would be greatly helpful if the reviewer could give a pointer to the similar investigations conducted in other tasks, as we would be glad to include comparisons with them.

---

### Meta-Review · Area_Chair_fsYT · 2023-09-18

**Recommendation:** 3

**Metareview:**

This paper proposes a novel prompt design strategy for text-to-SQL with in-context learning. The proposed approach first retrieves demonstrations for a test instance by generating preliminary SQL query and both diversity and similarity between demonstrations and the test instance are considered. This work introduces approaches to include schema-related knowledge in prompts, and ensemble approaches to aggregate results to tackle the sensitivity issue.

Combination of all aforementioned information, outperforms the SOTA of in-context learning methods and fine-tuned methods. The findings of this work could benefit the community for studying text-to-SQL.

---

### Decision · Program_Chairs · 2023-10-07

**Decision:**

Accept-Findings

**Comment:**

This paper proposes a novel prompt design strategy for text-to-SQL with in-context learning. The proposed approach first retrieves demonstrations for a test instance by generating preliminary SQL query and both diversity and similarity between demonstrations and the test instance are considered. This work introduces approaches to include schema-related knowledge in prompts, and ensemble approaches to aggregate results to tackle the sensitivity issue.

Combination of all aforementioned information, outperforms the SOTA of in-context learning methods and fine-tuned methods. The findings of this work could benefit the community for studying text-to-SQL.